# OpenReview forum: "From Trojan Horses to Castle Walls: Unveiling Bilateral Backdoor Effects in Diffusion Models"
_NeurIPS.cc/2023/Workshop/BUGS — NeurIPS 2023 BUGS Poster_

### Official Review · Reviewer_XXjz · 2023-10-26
**An interesting paper on backdoor diffusion models through the perspective of data poisoning**

**Rating:** 6
**Confidence:** 5

**Review:**

## Summary

The paper proposes a relaxed threat model to backdoor diffusion models (DM), with data poisoning attacks. The paper first poison the training dataset of DM, through injecting trigger pattern onto the training images, or misalign the image-text dataset. The paper provides insights on attack and defense sides with the backdoored DM. On the attack side, the backdoored DM is able to generate image with specific trigger pattern, or misaligned with the text prompt, while on the defense side, the backdoored DM can be used as a preprocessing step, where it is used to generate images with trigger pattern, and labelled correctly to the ground truth class.

## Strength

- The paper relaxed the assumption of threat models on launching backdoor attacks on DM
- The paper showed the backdoor attack effectiveness by injecting trigger onto DM training set and misalignment between image and text-prompt
- The paper showed an interesting view of how different is attacking DM compared to attacking normal image classification models
- The paper showed several possibilities of outcomes while backdooring DMs, which could be categorized as 4 categories, poisoned generation + misaligned image-text (G1), poisoned generation only (G2), misaligned image-text (G3) and normal generation (G4)
- Empirical results showed that backdoor DMs with data poisoning is feasible, allowing misaligned text-image generations and poisoned image generations

## Weaknesses

- Defending using backdoor DM is counter-intuitive, where the defender assumed that they would know what form of attack will come to them. For example, in line 210-212, the paper mentioned that backdoored DM could be utilized to generate poisoned generations with correct ground truth labels, which means that the defender would already know the incoming attack is BadNet, and they train a backdoor DM with BadNet. The underlying assumption for this is too strong, as it is impossible for the defender to identify what type of attack that will come to them.
- Defending with backdoor DM is also unreasonable, as clean DMs will generate artifacts along with the outputs, thus enhancing the image classifier’s robustness. The motivation of using backdoor DM for defend is unclear, and backdoor DM might create a plausible scenario for clean-label attack.

---

### Official Review · Reviewer_EFyV · 2023-10-26
**'The paper explores BadNet-like Trojans in diffusion models (DMs). It investigates the possibility of creating backdoor attacks in DMs by following a 'BadNets' behavior and constrains. The paper finds that BadNet can be adapted to DMs and provides several empirical findings to help protect the model.**

**Rating:** 5
**Confidence:** 3

**Review:**

Strengthens:

- Good writing.
- The paper provides some interesting empirical findings that may help understand Trojans in DMs.

Weakness:

- It's hard for me to understand why this paper claims that BadNet-like attacks would be more practical than existing attacks. I don't really see why changing the loss function during training of the poisoned model would be less feasible (put more constraints) than just changing the data and the training behavior. Both BadNet-like attacks and mentioned existing attacks against DMs fall into the category of attacks that require somehow controlling the training of the model and then distributing the compromised model. Therefore, changing or not changing the loss function and some other distributions doesn't seem to matter when it's completely under the attacker's control. Instead, changes specific to DMs make the general backdoor attacks perform better in specific DMs domains.
- Lack of technical contribution. It seems the paper only reuses BadNet techniques without solving new challenges.
- Lack of comparison with existing Trojans in DMs.
- Some findings are already discussed in the existing paper. For example, "Second, training image 54 classifiers with generated images from backdoored DMs before the phase transition can effectively 55 mitigate backdoor attacks." is similar to adversarial training.

---

### Decision · Program_Chairs · 2023-10-28

**Decision:**

Accept (Poster)

**Comment:**

Thank you for submitting to our workshop. Though the reviewers have concerns regarding the claims in this work, we believe backdoors in Diffusion Models is a direction worth studying and thus recommend acceptance.